# The Relationship of Socioeconomic Factors and Substance Abuse Treatment Dropout

**DOI:** 10.3390/healthcare13040369

**Published:** 2025-02-10

**Authors:** Wenyu Zhang, Hui Wu

**Affiliations:** 1Independent Researcher, Jersey City, NJ 07302, USA; wenyu_zhang@alumni.brown.edu; 2Independent Researcher, Bayonne, NJ 07002, USA

**Keywords:** substance use disorders, dropout, odds ratio, socioeconomic factors, alcohol, marijuana, heroine

## Abstract

**Background**: Treatment dropout in substance use disorder (SUD) programs poses a significant challenge to achieving successful outcomes and leads to legal and financial issues. Socioeconomic factors have been identified as key contributors to treatment attrition; yet, the specific impact of patients’ socioeconomic conditions remains underexplored. The purpose of this study is to examine the relationship between socioeconomic factors and SUD treatment dropout. **Methods**: We conducted a retrospective analysis of socioeconomic factors associated with treatment dropout among individuals with alcohol, marijuana, and heroin substance abuse. Logistic regression was used to examine the association between patients’ socioeconomic factors and treatment dropout. Adjusted odds ratios were calculated to quantify the strength of these associations. **Results**: Our findings demonstrate that demographic factors and financial status, including age (12–19 years), Black or African American race, and reliance on public assistance, correlate with an increased likelihood of treatment dropout. Black or African American patients receiving public assistance exhibit elevated dropout rates in ambulatory services, while patients of other single races without private insurance show higher dropout rates in detox services. Individuals aged 18–49 who are not part of the labor force have increased dropout rates in rehab services. Interestingly, patients in dependent living situations, who pay for services through private insurance or receive them at no charge, experience lower dropout rates as the number of arrests increases. Conversely, independently living patients with prior SUD treatments have higher dropout rates compared to those undergoing treatment for the first time. **Conclusions**: This study underscores the critical importance of addressing financial barriers to treatment access and retention in order to improve outcomes for individuals with substance use disorders (SUDs). Targeted interventions that support economically disadvantaged populations are essential for reducing treatment dropout rates and enhancing the effectiveness of SUD treatment programs.

## 1. Introduction

Substance use disorders (SUDs) affect an estimated 271 million people, approximately 5% of global population [1,2]. SUDs have extensive influence on individuals, families, communities, and societies in many ways. SUDs concurrently precipitate considerable familial distress and community issues and impose a substantial economic burden on society at large [3,4,5,6,7,8]. Also, SUDs exhibit a pervasive prevalence within the general population, engendering pronounced risks to both individual health and well-being. The expenses related to drug overdose constitute a critical portion of healthcare costs, with the rates of hospitalization due to SUDs having increased 55% in the United States in the past decade [9]. As the incidence of substance abuse continues to escalate, it becomes imperative to understand the underlying factors contributing to substance abuse [10].

A previous report showed that nearly 50% of patients drop out of substance treatment programs [11]. Dropping out of treatment is a significant problem since it directly relates to the likelihood of relapse or resuming substance use and causes legal and financial challenges [12,13,14]. Treatment dropout carries a substantial cost to society concerning HIV spread, increasing crime rate, and inflicting considerable emotional distress on families. Previous studies have identified that certain social demographic factors are risk factors for substance treatment dropout. José J. López-Goñi et al. showed that younger age and male gender present greater risk of dropping out of treatment [15]. King AC et al. showed that female gender and being African American are high risk predictors for early treatment dropout [16]. Ball SA et al. reported that demographic factors and patient personality and motivation are strongly associated with dropping out of an outpatient treatment program [17]. Martinez–Raga et al. reported that unplanned treatment discharge from inpatient alcohol detoxification is more likely to occur in younger clients [18]. Maglione et al. found that patients undergoing outpatient treatment for methamphetamine use who are older are more likely to complete the treatment while men are more likely to drop out [19]. López–Goñi et al. studied drug-dependent patients in an outpatient setting and found that patients without employment have a significantly higher dropout rate [20]. Perron et al. discovered that legal coercion reduces dropout risk in short-term residential treatment [21]. Sayre et al. found that in an outpatient cocaine treatment program, high dropout rate is closely related to marital status, low education, and being female [22].

In this article, we aim to study a large patient population to determine the average rates of treatment dropouts and examine the relationship between socioeconomic factors and SUD treatment dropout. This study utilizes data from the SAMHSA survey conducted between 2017 and 2020, which encompasses a wide range of patients. We focus on clients who are receiving various treatment services, including detoxification, rehabilitation, and ambulatory care and who have used any of the following three substances: alcohol, marijuana, or heroin. Additionally, our analysis of the overall population is repeated for subpopulations based on the type of service received and their prior arrest records or previous SUD treatments.

## 2. Methods

### 2.1. Data Source

This study uses the US Substance Abuse and Mental Health Services Administration’s (SAMHSA) Treatment Episode Dataset-Discharge (TEDS-D) from the year 2017 to 2020 [23]. This dataset is open access and thus open to the general public. It contains data collected by state health systems, and is the only national dataset on publicly funded SUD treatments. It records admissions and discharges of substance use clients aged 12 years or older into state-licensed or certified treatment centers that receive federal funding across the United States and jurisdictions. TEDS-D collects the demographics and substance use characteristics information of patients at discharge, not at admission. TEDS-D data count records by admission, not by client, so patients who were admitted multiple times were entered multiple times into the sample. The dataset only reports the primary, secondary, and tertiary substances used by a patient and does not report all drugs the patient may take.

### 2.2. Study Sample

The dataset classifies the discharge reason of a patient discharge record into seven categories: treatment completed, dropped out of treatment, terminated by facility, transferred to another treatment program or facility, incarcerated, death, and other. Since “other” is considered unknown and only represents 4% records, casewise deletion was utilized to remove those records listed with “other” as the discharge reason.

In addition, this study looks at patients who abuse at least one of the following three substances: alcohol, marijuana, or heroine. These are the three dominant types of drugs in the dataset. That means that a client is taken into account if their primary, secondary, or tertiary drug at admission falls in this group.

This study focuses on socioeconomic factors of the patients. Therefore, we look at the following categorical variables in the dataset: age, gender, race, marital status, level of education, employment status, veteran status, living arrangements, source of income, health insurance, and primary payment source. For each variable, the dataset includes a category for missing or unknown cases. When we look at the relationship between dropout and these variables, clients with missing information are removed from the analysis.

### 2.3. Measures

Records of patients’ substance treatment dropout were derived directly from the SAMHSA dataset, while records of patients who did not drop out of treatment were derived from either the treatment completed, terminated by facility, transferred to another treatment program or facility, incarcerated, or death categories. These different treatment services were categorized into three groups, including

detoxification: 24-h-per-day medical acute care services in a hospital or non-hospital setting for detoxification of persons with severe medical complications associated with withdrawal;rehabilitation/residential: non-acute care in a setting with treatment services for alcohol and other drug use and dependency;ambulatory: ambulatory treatment services including individual, family, and/or group services and may include pharmacological therapies.

To understand treatment dropout in different social demographic populations, we derived independent factors including age, gender, race, martial status, education, primary income, employment status, living arrangement, health insurance, primary payment type, and veteran status from the SAMHSA dataset. Compared with the original dataset, we regrouped the categories of variables Age and Race to avoid any categories with low populations. Age 12–14 and age 15–17 were grouped into age 12–17 to better represent the teenager age group. Age 18–20, 21–24, and 25–29 were grouped as age 18–29 to avoid low record counts in granular groups, and age 18–29 can also better represent the emerging adults group. Age 30–34, 35–39, 40–44, and 45–49 were grouped as age 30–49, which represents early middle adulthood, the period when people start families, careers, and relationships. Age 50–54 and 55–64 were grouped as age 50–64 to represent middle adulthood, the period of time of career peak and caring for families. Age over 65 was kept as its own age group to represent senior adulthood or senior people when they reach retirement. Indigenous groups, e.g., Native Hawaiian, American Indian, and Pacific Islander, were grouped as one race group since indigenous groups have their own unique languages, cultures, and spiritual beliefs and distinguish themselves from other races. In addition, for all variables we studied, we removed the samples where the entries on these variables are missing or unknown. Casewise deletion was utilized to remove those records, such as “other” discharge reason.

Age was categorized as age 12–17, age 18–29, age 30–49, age 50–64, and age over 65.Gender was categorized as male and female.Race was categorized as Asian, Black or African American, Native Hawaiian/American Indian/Pacific Islander, white, other single race, and more than two races. Native Hawaiian/American Indian/Pacific Islander was grouped from Alaska Native, American Indian (other than Alaska Native), Asian or Pacific Islander, Native Hawaiian, or Other Pacific Islander. Other single race was used when the patient was not identified in any of the categories above or when their origin group, due to local customs, was regarded as a distinct racial class from the categories listed above. “Two or more races” was used when the state data system allowed for multiple race selection and more than one race was indicated.Core-based statistical area (CBSA) is a geographic region comprising a central urban area and the surrounding counties that are socially and economically connected to the core.Marital status was categorized into married, separated, divorced (or widowed), and never married.Education was categorized as less than grade 8, grades 9–11, grade 12 (or GED), 1–3 years of college, and 4 years of college and BA/BS.Source of income was the primary income of a patient, categorized as wages, public assistance, retirement/pension/disability, other, and none income.Employment types were categorized as full-time, part-time, unemployed, and not in labor force. Unemployed patients were defined as those who were looking for work during the past 30 days or on layoff from a job. “Not in labor force” indicated patients who were not looking for work during the past 30 days or were a student, a homemaker, disabled, retired, or an inmate of an institution.Living arrangement was categorized into homeless, dependent living, and independent living. Dependent living was defined as a condition when patients were living in a supervised setting such as a residential institution, halfway house, or group home, and children (under age 18) living with parents, relatives, or guardians or (substance use clients only) in foster care. Independent living was defined as a condition when patients were living alone or with others in a private residence and were capable of self-care. Independent living also included adult children (age 18 and over) living with parents and adolescents living independently. Also, it included clients who lived independently with case management or supported housing support.Health insurance was categorized into private insurance, Medicare, Medicaid, or none insurance. Private insurance included private insurance, blue cross/blue shield, and HMO.Payment source was categorized into self-pay, private insurance, Medicare, Medicaid, other government payments, no charge, or others. Treatment payment with private insurance included Blue Cross/Blue Shield, other health insurance, and worker’s compensation. Treatment payment with no charge included free treatment, charity-supported treatment, special research, or teaching-purposes treatment.Veteran status was categorized as veteran or non-veteran.

### 2.4. Statistical Analysis

Cramer’s V measurement was employed to assess associations between socioeconomic factors and treatment dropout probability. Cramer’s V is the most commonly used strength test for Chi-square independence measurements. It shows how strongly two categorical fields are associated [24].

Bi-variate analyses were conducted to examine the unadjusted associations between each independent socioeconomic factor and the outcome variable: treatment dropout. An unadjusted logistic regression model was initially fitted to estimate the unadjusted odds ratios (OR) for each independent variable.

We also conducted a multivariate analysis with regression model to understand the adjusted odds ratio of contribution to treatment dropout. To figure out a variable’s different impacts on treatment dropout in different treatment services, we split the patient samples into 3 groups—ambulatory service, detoxification service (“detox” in this article), and rehabilitation/residential service (“rehab” in this article)—and conducted multivariate analysis to capture the adjusted odds ratio of dropout events. A 95% confidence interval of adjusted odds ratio was computed in order to determine whether a variable or a category of variables was significant.

In addition, we studied various populations depending on their previous arrests in the past 30 days and prior treatments. The same multivariate analysis was repeated for these select sub-populations.

## 3. Results

### 3.1. Correlation Between Socioeconomic Observables and Dropout

Cramer’s V analysis was employed to understand the correlation between different socioeconomic factors and treatment dropout. A total of 912,832 patient admission records received treatments, among which 622,965 were treated with ambulatory services, 134,872 were treated using detox services, and 154,995 were treated using rehab services. Figure 1 gives an example of the correlation between gender and treatment dropout.

Geographic factors including state and core-based statistical area (CBSA) are highly correlated with treatment dropout (Table 1). The geographic importance also extends to different service types, with the highest correlation in ambulatory services and lowest correlation in rehabilitation services (Table 1).

### 3.2. Descriptive Statistics

A total of 912,832 patient records were included in the study, with 220,369 (24.14%) dropping out of treatments and 692,463 (75.86%) not dropping out of treatments. The majority of patients were aged 30–49 (11.78% drop out, 37.59% not drop out), male (15.08% drop out, 48.78% not drop out), white (17.04% drop out, 54.53% not drop out), not Hispanic or Latino (21.47% drop out, 68.08% did not drop out), never married (16.31% drop out, 48.74% not drop out), education up to grade 12 (12.03% drop out, 38.18% not drop out), without income (10.37% drop out, 33.86% not drop out), unemployed (9.01% drop out, 30.95% not drop out), independently living (15.88% drop out, 45.86% not drop out), had Medicaid health insurance (11.09% drop out, 33.83% not drop out), used Medicaid as their primary treatment payment (10% drop out, 31.15% not drop out), and had not served in military (23.53% drop out, 73.55% not drop out). The descriptive stats were also separated into 3 service types (ambulatory, detox, and rehab) and are presented in Table 2.

### 3.3. Substance Treatment Dropout by Service Types

The unadjusted odds ratio was utilized to understand the relative odds of treatment dropout for different categories of independent variables. Patients with age 12–17 (*OR* = 1.26 [1.24–1.27]), with other single race (*OR* = 1.33 [1.31–1.1.33]), Hispanic or Latino origin (*OR* = 1.34 [1.33–1.35]), with public assistance as their primary income (*OR* = 1.29 [1.28–1.30]), and not in the labor force (*OR* = 1.22 [1.21–1.23]), and receiving treatments with no charge (*OR* = 2.11 [2.07–2.15]) had higher odds of dropping out of substance treatments. Patients who were of Native Hawaiian/American Indian/Pacific Islander race (*OR* = 0.72 [0.70–0.72]), not Hispanic or Latino in origin (*OR* = 0.74 [0.73–0.74]), had private insurance, Blue Cross/Blue Shield, or HMO (*OR* = 0.73 [0.72–0.80]), and paid the treatment with self–pay (*OR* = 0.67 [0.65–0.68]) were less likely to drop out of treatments.

Table 3 shows the results of the adjusted odds ratio and its 95% confidence interval using the logistic regression method for the entire population and for patients of three service types. For each variable, there is one category which is considered as reference when conducting regression so that all odds ratios will be a comparison to the reference group. For example, for “Age”, the odds ratio of categories except for the 12–17 group shows the probability of other patients dropping out compared to that of the 12–17 age group.

For the entire population studied, patients aged equal to or greater than 65 had a much lower chance of dropping out of treatment (OR=0.59) compared to other age groups. Black or African American (OR=1.56) race, patients with public assistance as their primary income (OR=1.52), people with Medicaid insurance (OR=1.50) or without health insurance (OR=1.56), people who paid with private insurance (OR=1.74) or Medicare (OR=1.65), those with no charge (OR=2.54), and those with other payment sources (OR=1.99) had a significantly higher probability of dropping out.

Among the three groups of patients in different types of services, people aged 50 years or older had a much lower dropout rate with detox services (OR≤0.52), but this group did not have an obviously low dropout rate in other types of services. The Black or African American race group had a high dropout rate with ambulatory services. The “other single race” group had a high dropout rate with detox services but a rather low dropout rate with rehab services. In the two or more races and white group, people undergoing detox services had a higher dropout rate (OR≥1.51). People with an education of 4 years college or beyond had a lower chance of dropping out (OR≤0.55) of detox services. People with public assistance as their primary income had a higher dropout rate with ambulatory services (OR=1.77). People not in the labor force had a higher dropout rate with rehab services (OR=1.62). People having an independent living arrangement had a higher dropout rate with detox services (OR=1.45). For the detox service group, patients with Medicaid, Medicare, or no insurance all had high dropout rates (OR>2.7), while only patients with no insurance had a high dropout rate in the ambulatory group, and only the Medicare insurance group had a high dropout in the rehab service group. Regarding payment sources, people paying with private insurance or other sources had a high dropout rate in both the ambulatory group and detox group. People paying with other government payment methods had high dropout rates in all the service groups.

Overall, for all individual service type groups and the entire population, gender, marital status, and veteran status did not play an important role in probability of dropping out.

### 3.4. Arrests Prior to Admission’s Influence on Patient Treatment Dropout

Table 4 shows the adjusted odds ratio and confidence interval results for various sub-populations depending on the number of prior arrests of the patient. The population within the patient group was 844,674, 56,477, and 8432 for prior arrests numbering 0, 1, and 2 or more.

Among the three sub-populations, it was observed that people aged 65 or above always had a lower dropout rate compared with other age groups (OR<0.7). Black or African American patients always had a high dropout rate (OR>1.37). Oppositely, Native Hawaiian/American Indian/Pacific Islander patients had a lower chance of dropping out (OR<0.7) for patients with 2 or 1 prior arrests. In the no prior arrest group, people paying with public assistance had a higher dropout rate. People without health insurance had high dropout rates no matter how many prior arrests they had (OR≥1.46). For patients with no prior arrests, people using Medicaid insurance had high dropout rates (OR=1.52). Regarding primary payment source, people paying with private insurance or Medicare, with other sources, or with no charge had high dropout rates in the no prior arrests group, while for people with two or more prior arrests, those paying with Medicare or other sources had a higher chance of dropping out (OR≥1.36).

### 3.5. Previous Substance Use Treatment Episodes’ Influence on Patient Treatment Dropout

Table 5 shows the adjusted odds ratio and its 95% confidence interval for patients with different prior treatment experience. The prevalence of no prior treatment in the total population was 288,702, and the prevalence of patients having one or more prior treatment in the total population was 576,269.

It was observed that for both patients with prior treatments or without prior treatments, those who were aged 65 or above had a lower dropout rate (OR<0.7). Patients whose primary income was public assistance, who did not have health insurance, who payed with private insurance, Medicare, other sources, or who did not pay had higher dropout rates (OR>1.4). Race impacted differently on dropout performance for patients with or without prior treatments. The Native Hawaiian/American Indian/Pacific Islander group had lower dropout rates among patients who had prior treatments, while the “other single race” group had a lower dropout rate among patients who did not have prior treatments. In addition, patients who were in independent living arrangements or used Medicaid insurance had a higher dropout rate if they had prior treatments (OR≥1.47).

## 4. Discussion

This study focused on the population in the U.S. from 2017 to 2020 who abused alcohol, marijuana, or heroin. We examined their socioeconomic factors to understand the relationship between these factors and treatment dropout. Our study presents several strengths, including a comprehensive analysis of demographic and socioeconomic association with treatment dropout among individuals with substance use disorders. By employing logistic regression analysis, we identified significant predictors of dropout probability both in the overall population and across different service types, enhancing our understanding of treatment retention dynamics. Additionally, the inclusion of a diverse sample strengthened the generalizability of our findings, as we examined patients of various age groups, racial backgrounds, geographical locations, and socioeconomic statuses. Further analysis was conducted on dropout probability in subpopulations, focusing on patients with varying histories of prior arrests or previous SUD treatments.

Substance abuse treatment has a dropout rate of approximately 24% among patients. The 24% dropout rate observed in our dataset aligns the findings of two U.S.-based studies, which reported rates of 21% and 25% [25,26]. However, this prevalence stands notably lower than the dropout rates reported in other studies, ranging from approximately 50% to 88% [27,28,29]. These discrepancies may be attributed to variations in how dropout is defined as well as differences in sample sizes—our study included 912,832 admission records, whereas prior studies involved only a few hundred.

To investigate the determinants of dropout, we evaluated discharged patients from treatment facilities to understand the causes of discontinuation and to identify connections between various socioeconomic factors and treatment attrition. Our analysis reveals a strong correlation between geographical factors, such as state and CBSA (Core-Based Statistical Area) regions, and patient dropout. This finding is consistent with previous studies [30,31]. Urban areas and rural areas may have different access to treatment. Compared to urban areas, people living in rural areas generally have significantly less access to healthcare and treatments due to a shortage of treatment facilities, longer travel distances to reach medical facilities, and limited availability, often leading to delayed or forgone medical treatment; this can result in poorer health outcomes for rural populations.

Our study shows that in both the overall population and subpopulations considering prior arrests and prior treatments, individuals over 65 consistently exhibit a lower probability of dropout compared to other age groups. This result is consistent with the findings of José J. López-Goñi et al. [15] and McKellar et al. [32], which indicate that younger individuals have a higher risk of dropping out. This suggests that older individuals may demonstrate greater adherence to treatment regimens, potentially due to factors such as increased health awareness or a stronger support network.

In the various populations we investigated, we observed that Black or African American individuals consistently have a higher probability of dropout compared to other racial groups. A report by Saloner and Cook found a similar trend, indicating that Black, Hispanic, and Native American individuals are less likely than white individuals to complete treatment for substance use disorders. The authors attributed these disparities to factors such as unemployment, housing instability, and other underlying issues [33]. The study by Mennis and Stahler also showed that African Americans are less likely to complete treatment compared to whites in urban outpatient substance use disorder programs, primarily due to social isolation [34].

SUD treatment dropout is closely related to patients’ financial situation. It has been observed that patients whose primary source of income is public assistance are more likely to discontinue treatment. Additionally, those with Medicaid insurance or no insurance at all have a higher likelihood of dropping out. Our findings align with a previous study highlighting the significant impact of patients’ financial problems on substance abuse treatment dropout [35]. Cook et al. found that patients with Medicaid insurance are more likely to enroll in SUD treatments; however, their study did not investigate dropout events [36].

Regarding clients in different treatment services, we found that Black or African American patients receiving public assistance exhibit elevated dropout rates in ambulatory services. In contrast, patients of other racial backgrounds without private insurance show higher dropout rates in detox services. Individuals aged 18 to 49 who are not in the labor force display increased dropout rates in rehab services. Interestingly, patients who identify as Native Hawaiian, American Indian, Pacific Islander, or another single race tend to have a lower treatment dropout rate as number of arrests increases. In addition, prior arrests and previous SUD treatments affect dropout rates differently. Patients in dependent living situations who pay for services through private insurance or receive them at no charge have lower dropout rates as the number of arrests increases. Conversely, independently living patients with prior SUD treatments exhibit higher dropout rates compared to those undergoing treatment for the first time.

Our study may be limited by its reliance on secondary data collected from treatment facilities, which could introduce data limitations and selection bias, particularly concerning the accuracy and completeness of patient records. While we identified significant predictors of treatment dropout, the cross-sectional nature of the data restricts our ability to infer causality. Future research utilizing longitudinal study designs could provide deeper insights into the temporal relationships between demographic factors, service utilization, and treatment outcomes.

Despite controlling for various demographic and socioeconomic variables, our analysis may not account for all potential confounders influencing treatment dropout. Factors such as social support networks, treatment engagement, and comorbid psychiatric conditions warrant further investigation in future studies. The treatment environment, for example, whether adjustments can be made to accommodate specific patient needs, is important to the patient’s decision to remain in a treatment [11]. Poorer psychiatric functioning [22], anxiety sensitivity [26], and having personality disorders [18] are related to dropout risks, too. Additionally, it is likely that some factors we studied are interdependent; for example, a patient’s primary source of income may influence the type of health insurance they use and their payment methods.

In conclusion, this study underscores the strong association between patients’ socioeconomic factors, demographic characteristics, and treatment dropout among individuals with substance use disorders (SUDs). Specifically, we found that geographic factors including state and core-based statistical area (CBSA) are highly correlated with treatment dropout. With the entire population studied, patients aged equal to or greater than 65 have a much lower chance of dropping out the treatment compared with other age groups. Additionally, Black or African American patients who rely on public assistance as their primary source of income, as well as those with Medicaid insurance or lacking coverage, are at a heightened risk of discontinuing treatment. Native American/American Indian/Pacific Islander groups tend to have a lower dropout rate when they had arrests prior to admission or patients have received prior SUD treatments. These findings highlight an urgent need for targeted interventions that address financial barriers to treatment access and retention, ultimately aiming to improve outcomes for individuals with SUDs.

## Figures and Tables

**Figure 1 healthcare-13-00369-f001:**
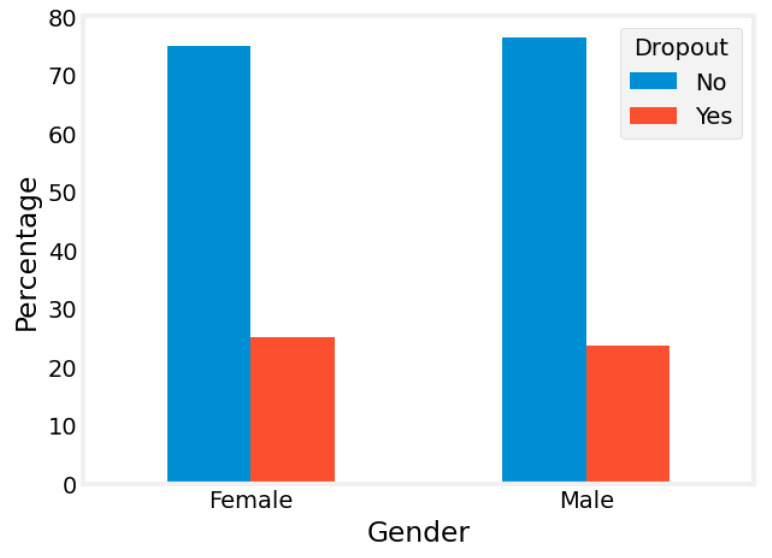
Correlation between gender and treatment dropout.

**Table 1 healthcare-13-00369-t001:** Cramer’s V correlation between socioeconomic factors and treatment dropout.

	All_Services	Ambulatory	Detox	Rehab
CBSA	0.29	0.38	0.28	0.25
State	0.25	0.35	0.27	0.19
Health insurance	0.05	0.05	0.11	0.07
Payment source	0.10	0.11	0.19	0.10
Age	0.05	0.04	0.11	0.04
Gender	0.02	0.01	0.00	0.01
Race	0.05	0.08	0.06	0.04
Martial status	0.03	0.03	0.05	0.04
Education	0.04	0.04	0.05	0.06
Primary income	0.05	0.10	0.05	0.07
Veteran	0.01	0.01	0.02	0.01
Living arrangement	0.02	0.02	0.05	0.06
Employment	0.05	0.09	0.04	0.07

**Table 2 healthcare-13-00369-t002:** Descriptive statistics of socioeconomic factors.

	Inclusive		Ambulatory		Detox		Rehab	
Dropout	Yes	No	Yes	No	Yes	No	Yes	No
**Age**								
Age 12–17	10,437 (1.14%)	27,635 (3.03%)	8976 (1.44%)	21,074 (3.38%)	2 (0.00%)	13 (0.01%)	1459 (0.94%)	6548 (4.22%)
Age 18–29	70,888 (7.77%)	196,737 (21.55%)	54,211 (8.7%)	140,474 (22.55%)	6792 (5.04%)	22,426 (16.63%)	9885 (6.38%)	33,837 (21.83%)
Age 30–49	107,490 (11.78%)	343,090 (37.59%)	78,435 (12.59%)	224,945 (36.11%)	13,127 (9.73%)	56,638 (41.99%)	15,928 (10.28%)	61,507 (39.68%)
Age 50–64	29,685 (3.25%)	116,571 (12.77%)	22,420 (3.60%)	65,703 (10.55%)	3218 (2.39%)	30,372 (22.52%)	4047 (2.61%)	20,496 (13.22%)
Age 65+	1869 (0.20%)	8430 (0.92%)	1560 (0.25%)	5167 (0.83%)	134 (0.10%)	2150 (1.59%)	175 (0.11%)	1113 (0.72%)
**Gender**								
Male	137,625 (15.08%)	445,320 (48.78%)	102,555 (16.46%)	284,110 (45.61%)	16,065 (11.91%)	83,078 (61.6%)	19,005 (12.26%)	78,132 (50.41%)
Female	82,744 (9.06%)	247,143 (27.07%)	63,047 (10.12%)	173,253 (27.81%)	7208 (5.34%)	28,521 (21.15%)	12,489 (8.06%)	45,369 (29.27%)
**Race**								
Asian	1159 (0.13%)	4891 (0.54%)	839 (0.13%)	3361 (0.54%)	103 (0.08%)	671 (0.5%)	217 (0.14%)	859 (0.55%)
Black or African American	48,932 (5.36%)	120,690 (13.22%)	39,758 (6.38%)	80,382 (12.90%)	3306 (2.45%)	19,086 (14.15%)	5868 (3.79%)	21,222 (13.69%)
Native Hawaiian/American Indian/Pacific Islander	6062 (0.66%)	38,856 (4.26%)	4155 (0.67%)	28,382 (4.56%)	641 (0.48%)	5612 (4.16%)	1266 (0.82%)	4862 (3.14%)
Other single race	4810 (0.53%)	19,034 (2.09%)	3797 (0.61%)	14,182 (2.28%)	497 (0.37%)	1694 (1.26%)	516 (0.33%)	3158 (2.04%)
Two or more races	3843 (0.42%)	11,188 (1.23%)	2827 (0.45%)	7428 (1.19%)	423 (0.31%)	1682 (1.25%)	593 (0.38%)	2078 (1.34%)
White	155,563 (17.04%)	497,804 (54.53%)	114,226 (18.34%)	323,628 (51.95%)	18,303 (13.57%)	82,854 (61.43%)	23,034 (14.86%)	91,322 (58.92%)
**Marital status**								
Never married	148,920 (16.31%)	444,881 (48.74%)	111,486 (17.90%)	289,133 (46.41%)	15,776 (11.7%)	73,315 (54.36%)	21,658 (13.97%)	82,433 (53.18%)
Married	26,784 (2.93%)	88,453 (9.69%)	21,523 (3.45%)	64,409 (10.34%)	2365 (1.75%)	10,964 (8.13%)	2896 (1.87%)	13,080 (8.44%)
Separated	14,253 (1.56%)	46,309 (5.07%)	10,161 (1.63%)	31,278 (5.02%)	1695 (1.26%)	6390 (4.74%)	2397 (1.55%)	8641 (5.58%)
Divorced	30,412 (3.33%)	112,820 (12.36%)	22,432 (3.60%)	72,543 (11.64%)	3437 (2.55%)	20,930 (15.52%)	4543 (2.93%)	19,347 (12.48%)
**Education**								
Less than Grade 8	10,588 (1.16%)	36,342 (3.98%)	7782 (1.25%)	24,728 (3.97%)	909 (0.67%)	4187 (3.10%)	1897 (1.22%)	7427 (4.79%)
Grades 9 to 11	52,466 (5.75%)	139,538 (15.29%)	39,620 (6.36%)	95,727 (15.37%)	4861 (3.60%)	17,788 (13.19%)	7985 (5.15%)	26,023 (16.79%)
Grade 12 (or GED)	109,830 (12.03%)	348,528 (38.18%)	81,571 (13.09%)	226,279 (36.32%)	12,827 (9.51%)	60,879 (45.14%)	15,432 (9.96%)	61,370 (39.59%)
1–3 years of college	37,232 (4.08%)	125,283 (13.72%)	28,552 (4.58%)	83,645 (13.43%)	3705 (2.75%)	20,519 (15.21%)	4975 (3.21%)	21,119 (13.63%)
4 years of college, BA/BS	10,253 (1.12%)	42,772 (4.69%)	8077 (1.30%)	26,984 (4.33%)	971 (0.72%)	8226 (6.10%)	1205 (0.78%)	7562 (4.88%)
**Source of income**								
Wages	66,157 (7.25%)	221,160 (24.23%)	58,830 (9.44%)	179,481 (28.81%)	3660 (2.71%)	21,169 (15.7%)	3667 (2.37%)	20,510 (13.23%)
Public assistance	16,262 (1.78%)	36,704 (4.02%)	13,463 (2.16%)	23,859 (3.83%)	971 (0.72%)	5759 (4.27%)	1828 (1.18%)	7086 (4.57%)
Retirement/pension/disability	16,972 (1.86%)	52,436 (5.74%)	13,851 (2.22%)	34,548 (5.55%)	1047 (0.78%)	9358 (6.94%)	2074 (1.34%)	8530 (5.5%)
Other	26,329 (2.88%)	73,036 (8.00%)	20,377 (3.27%)	50,233 (8.06%)	2883 (2.14%)	9450 (7.01%)	3069 (1.98%)	13,353 (8.62%)
None	94,649 (10.37%)	309,127 (33.86%)	59,081 (9.48%)	169,242 (27.17%)	14,712 (10.91%)	65,863 (48.83%)	20,856 (13.46%)	74,022 (47.76%)
**Employment**								
Full-time	44,581 (4.88%)	154,163 (16.89%)	40,245 (6.46%)	127,269 (20.43%)	2497 (1.85%)	14,487 (10.74%)	1839 (1.19%)	12,407 (8.00%)
Part-time	18,392 (2.01%)	51,751 (5.67%)	16,729 (2.69%)	42,248 (6.78%)	916 (0.68%)	5773 (4.28%)	747 (0.48%)	3730 (2.41%)
Unemployed	82,239 (9.01%)	282,542 (30.95%)	59,140 (9.49%)	174,243 (27.97%)	10,670 (7.91%)	55,041 (40.81%)	12,429 (8.02%)	53,258 (34.36%)
Not in labor force	75,157 (8.23%)	204,007 (22.35%)	49,488 (7.94%)	113,603 (18.24%)	9190 (6.81%)	36,298 (26.91%)	16,479 (10.63%)	54,106 (34.91%)
**Living arrangement**								
Homeless	28,993 (3.18%)	117,042 (12.82%)	14,235 (2.29%)	47,783 (7.67%)	6679 (4.95%)	42,462 (31.48%)	8079 (5.21%)	26,797 (17.29%)
Dependent living	46,436 (5.09%)	156,802 (17.18%)	31,822 (5.11%)	93,606 (15.03%)	3878 (2.88%)	19,002 (14.09%)	10,736 (6.93%)	44,194 (28.51%)
Independent living	144,940 (15.88%)	418,619 (45.86%)	119,545 (19.19%)	315,974 (50.72%)	12,716 (9.43%)	50,135 (37.17%)	12,679 (8.18%)	52,510 (33.88%)
**Health insurance**								
Private insurance	20,804 (2.28%)	82,321 (9.02%)	17,167 (2.76%)	53,589 (8.60%)	1215 (0.9%)	12,807 (9.5%)	2422 (1.56%)	15,925 (10.27%)
Medicaid	101,222 (11.09%)	308,789 (33.83%)	74,971 (12.03%)	201,870 (32.40%)	9135 (6.77%)	43,794 (32.47%)	17,116 (11.04%)	63,125 (40.73%)
Medicare	9991 (1.09%)	45,343 (4.97%)	7603 (1.22%)	35,607 (5.72%)	812 (0.60%)	3831 (2.84%)	1576 (1.02%)	5905 (3.81%)
None	88,352 (9.68%)	256,010 (28.05%)	65,861 (10.57%)	166,297 (26.69%)	12,111 (8.98%)	51,167 (37.94%)	10,380 (6.70%)	38,546 (24.87%)
**Payment source**								
Self-pay	14,753 (1.62%)	65,635 (7.19%)	12,844 (2.06%)	51,355 (8.24%)	825 (0.61%)	9469 (7.02%)	1084 (0.7%)	4811 (3.10%)
Private insurance	17,029 (1.87%)	59,144 (6.48%)	14,242 (2.29%)	38,276 (6.14%)	1097 (0.81%)	8453 (6.27%)	1690 (1.09%)	12,415 (8.01%)
Medicare	1693 (0.19%)	5485 (0.6%)	1467 (0.24%)	3210 (0.52%)	51 (0.04%)	1595 (1.18%)	175 (0.11%)	680 (0.44%)
Medicaid	91,260 (10.0%)	284,317 (31.15%)	71,781 (11.52%)	187,104 (30.03%)	7538 (5.59%)	49,462 (36.67%)	11,941 (7.7%)	47,751 (30.81%)
Other government payments	58,860 (6.45%)	209,420 (22.94%)	36,625 (5.88%)	130,497 (20.95%)	9020 (6.69%)	30,249 (22.43%)	13,215 (8.53%)	48,674 (31.4%)
No charge	15,688 (1.72%)	23,966 (2.63%)	10,657 (1.71%)	14,990 (2.41%)	3154 (2.34%)	6183 (4.58%)	1877 (1.21%)	2793 (1.80%)
Other	21,086 (2.31%)	44,496 (4.87%)	17,986 (2.89%)	31,931 (5.13%)	1588 (1.18%)	6188 (4.59%)	1512 (0.98%)	6377 (4.11%)
**Veteran status**								
Yes	5625 (0.62%)	21,114 (2.31%)	4379 (0.7%)	13,726 (2.20%)	582 (0.43%)	4411 (3.27%)	664 (0.43%)	2977 (1.92%)
No	214,744 (23.53%)	671,349 (73.55%)	161,223 (25.88%)	443,637 (71.21%)	22,691 (16.82%)	107,188 (79.47%)	30,830 (19.89%)	120,524 (77.76%)

**Table 3 healthcare-13-00369-t003:** Adjusted odds ratio and 95% confidence interval with the whole population and three service types: ambulatory, detox, and rehab.

Variable	Inclusive	Ambulatory	Detox	Rehab
**Age**				
Age 12–17	1.00 (reference)	1.00 (reference)	1.00 (reference)	1.00 (reference)
Age 18–29	1.01 (0.98, 1.04)	0.95 (0.91, 0.98)	1.16 (0.24, 5.71)	1.62 (1.50, 1.75)
Age 30–49	0.89 (0.87, 0.92)	0.89 (0.85, 0.92)	0.93 (0.19, 4.56)	1.43 (1.33, 1.55)
Age 50–64	0.71 (0.68, 0.73)	0.83 (0.80, 0.87)	0.52 (0.11, 2.55)	1.08 (0.99, 1.17)
Age 65+	0.59 (0.56, 0.64)	0.72 (0.66, 0.77)	0.37 (0.07, 1.84)	0.89 (0.73, 1.08)
**Gender**				
Male	1.00 (reference)	1.00 (reference)	1.00 (reference)	1.00 (reference)
Female	1.03 (1.02, 1.04)	0.97 (0.96, 0.98)	1.14 (1.10, 1.18)	1.10 (1.06, 1.13)
**Race**				
Asian	1.00 (reference)	1.00 (reference)	1.00 (reference)	1.00 (reference)
Black or African American	1.56 (1.45, 1.68)	1.72 (1.58, 1.87)	1.23 (0.96, 1.58)	1.04 (0.88, 1.24)
Native Hawaiian/American Indian/ Pacific Islander	0.72 (0.66, 0.78)	0.68 (0.62, 0.75)	1.05 (0.80, 1.36)	1.01 (0.84, 1.21)
Other single race	0.85 (0.79, 0.93)	0.85 (0.77, 0.93)	2.10 (1.59, 2.76)	0.62 (0.51, 0.76)
Two or more races	1.25 (1.15, 1.36)	1.28 (1.16, 1.41)	1.54 (1.17, 2.02)	1.05 (0.86, 1.27)
White	1.26 (1.17, 1.35)	1.32 (1.21, 1.44)	1.51 (1.18, 1.93)	0.98 (0.83, 1.16)
**Marital status**				
Never married	1.00 (reference)	1.00 (reference)	1.00 (reference)	1.00 (reference)
Married	0.98 (0.96, 1.00)	0.92 (0.90, 0.94)	1.18 (1.11, 1.25)	1.06 (1.01, 1.12)
Separated	0.96 (0.94, 0.98)	0.88 (0.86, 0.91)	1.16 (1.09, 1.24)	1.10 (1.04, 1.16)
Divorced	0.94 (0.92, 0.96)	0.89 (0.87, 0.90)	1.04 (0.99, 1.09)	1.01 (0.97, 1.06)
**Education**				
Less than Grade 8	1.00 (reference)	1.00 (reference)	1.00 (reference)	1.00 (reference)
Grades 9 to 11	1.24 (1.20, 1.27)	1.27 (1.23, 1.31)	1.05 (0.96, 1.15)	1.07 (1.00, 1.14)
Grade 12 (or GED)	1.11 (1.08, 1.14)	1.19 (1.15, 1.23)	0.83 (0.76, 0.90)	0.87 (0.81, 0.92)
1–3 years of college	1.08 (1.05, 1.11)	1.17 (1.14, 1.22)	0.73 (0.66, 0.80)	0.85 (0.79, 0.91)
4 years of college, BA/BS	0.93 (0.90, 0.97)	1.09 (1.04, 1.13)	0.55 (0.49, 0.62)	0.64 (0.58, 0.70)
**Source of income**				
Wages	1.00 (reference)	1.00 (reference)	1.00 (reference)	1.00 (reference)
Public assistance	1.52 (1.48, 1.56)	1.77 (1.72, 1.83)	0.91 (0.81, 1.01)	1.00 (0.93, 1.09)
Retirement/pension/disability	1.15 (1.11, 1.18)	1.19 (1.15, 1.23)	0.75 (0.67, 0.84)	0.97 (0.90, 1.06)
Other	1.20 (1.17, 1.23)	1.23 (1.20, 1.27)	1.36 (1.24, 1.49)	0.97 (0.91, 1.04)
None	1.04 (1.02, 1.06)	1.06 (1.03, 1.08)	0.99 (0.91, 1.08)	1.07 (1.01, 1.14)
**Employment**				
Full-time	1.00 (reference)	1.00 (reference)	1.00 (reference)	1.00 (reference)
Part-time	1.15 (1.12, 1.18)	1.19 (1.16, 1.22)	0.94 (0.86, 1.03)	1.22 (1.10, 1.36)
Unemployed	0.96 (0.93, 0.98)	0.97 (0.95, 1.00)	1.05 (0.95, 1.15)	1.22 (1.13, 1.32)
Not in labor force	1.19 (1.16, 1.21)	1.22 (1.19, 1.26)	1.26 (1.14, 1.38)	1.62 (1.50, 1.76)
**Living arrangement**				
Homeless	1.00 (reference)	1.00 (reference)	1.00 (reference)	1.00 (reference)
Dependent living	1.07 (1.05, 1.09)	0.98 (0.95, 1.01)	0.98 (0.93, 1.03)	0.81 (0.78, 0.84)
Independent living	1.32 (1.30, 1.34)	1.15 (1.12, 1.18)	1.45 (1.39, 1.51)	0.87 (0.84, 0.90)
**Health insurance**				
Private insurance	1.00 (reference)	1.00 (reference)	1.00 (reference)	1.00 (reference)
Medicaid	1.50 (1.45, 1.55)	1.10 (1.06, 1.14)	3.25 (2.87, 3.68)	1.29 (1.19, 1.41)
Medicare	1.08 (1.04, 1.12)	0.82 (0.79, 0.86)	2.50 (2.18, 2.88)	1.36 (1.24, 1.50)
None	1.56 (1.52, 1.61)	1.47 (1.42, 1.52)	2.72 (2.41, 3.06)	1.12 (1.03, 1.22)
**Payment source**				
Self-pay	1.00 (reference)	1.00 (reference)	1.00 (reference)	1.00 (reference)
Private insurance	1.74 (1.68, 1.81)	2.00 (1.92, 2.08)	2.54 (2.21, 2.91)	0.76 (0.68, 0.84)
Medicare	1.65 (1.54, 1.77)	2.44 (2.25, 2.64)	0.52 (0.38, 0.71)	0.93 (0.75, 1.14)
Medicaid	1.26 (1.23, 1.30)	1.68 (1.63, 1.74)	1.27 (1.16, 1.40)	0.86 (0.79, 0.94)
Other government payments	1.18 (1.16, 1.21)	1.15 (1.12, 1.18)	2.53 (2.31, 2.77)	0.98 (0.90, 1.06)
No charge	2.54 (2.46, 2.62)	2.57 (2.48, 2.66)	4.23 (3.83, 4.66)	2.45 (2.21, 2.71)
Other	1.99 (1.94, 2.05)	2.19 (2.12, 2.25)	2.53 (2.28, 2.81)	0.98 (0.89, 1.08)
**Veteran status**				
Yes	1.00 (reference)	1.00 (reference)	1.00 (reference)	1.00 (reference)
No	1.06 (1.02, 1.09)	1.05 (1.01, 1.09)	1.05 (0.95, 1.17)	0.96 (0.87, 1.06)

**Table 4 healthcare-13-00369-t004:** Adjusted odds ratio and 95% confidence interval for patients grouped by number of arrests prior to patient admission.

Variable	ARRESTS = 0	ARRESTS = 1	ARRESTS = 2
**Age**			
Age 12–17	1.00 (reference)	1.00 (reference)	1.00 (reference)
Age 18–29	0.98 (0.95, 1.01)	1.12 (1.01, 1.25)	0.80 (0.59, 1.09)
Age 30–49	0.87 (0.84, 0.90)	0.95 (0.85, 1.06)	0.75 (0.55, 1.01)
Age 50–64	0.68 (0.66, 0.71)	0.76 (0.66, 0.86)	0.68 (0.48, 0.97)
Age 65+	0.58 (0.54, 0.62)	0.68 (0.47, 0.97)	0.66 (0.31, 1.41)
**Gender**			
Male	1.00 (reference)	1.00 (reference)	1.00 (reference)
Female	1.04 (1.03, 1.05)	0.98 (0.93, 1.03)	1.01 (0.89, 1.15)
**Race**			
Asian	1.00 (reference)	1.00 (reference)	1.00 (reference)
Black or African American	1.57 (1.46, 1.70)	1.74 (1.27, 2.37)	1.37 (0.58, 3.23)
Native Hawaiian/American Indian/Pacific Islander	0.74 (0.68, 0.80)	0.67 (0.48, 0.93)	0.46 (0.19, 1.12)
Other single race	0.90 (0.83, 0.98)	0.79 (0.56, 1.11)	0.39 (0.15, 1.00)
Two or more races	1.26 (1.16, 1.37)	1.36 (0.97, 1.92)	1.24 (0.49, 3.15)
White	1.27 (1.18, 1.37)	1.35 (0.99, 1.83)	1.04 (0.45, 2.43)
**Marital status**			
Never married	1.00 (reference)	1.00 (reference)	1.00 (reference)
Married	0.98 (0.97, 1.00)	0.99 (0.91, 1.06)	1.11 (0.92, 1.35)
Separated	0.96 (0.94, 0.99)	0.97 (0.88, 1.06)	1.17 (0.92, 1.48)
Divorced	0.93 (0.92, 0.95)	0.93 (0.87, 1.00)	0.87 (0.73, 1.04)
**Education**			
Less than Grade 8	1.00 (reference)	1.00 (reference)	1.00 (reference)
Grades 9 to 11	1.24 (1.20, 1.27)	1.27 (1.14, 1.41)	1.16 (0.88, 1.52)
Grade 12 (or GED)	1.11 (1.08, 1.14)	1.14 (1.03, 1.27)	0.98 (0.75, 1.28)
1–3 years of college	1.07 (1.03, 1.10)	1.19 (1.06, 1.34)	0.88 (0.66, 1.19)
4 years of college, BA/BS	0.94 (0.91, 0.98)	0.94 (0.80, 1.09)	0.77 (0.52, 1.14)
**Source of income**			
Wages	1.00 (reference)	1.00 (reference)	1.00 (reference)
Public assistance	1.55 (1.51, 1.60)	1.21 (1.07, 1.38)	1.27 (0.89, 1.82)
Retirement/pension/disability	1.15 (1.11, 1.18)	1.16 (1.02, 1.33)	1.32 (0.93, 1.85)
Other	1.20 (1.17, 1.23)	1.13 (1.02, 1.24)	1.24 (0.95, 1.60)
None	1.04 (1.02, 1.07)	0.91 (0.83, 0.99)	1.00 (0.80, 1.26)
**Employment**			
Full-time	1.00 (reference)	1.00 (reference)	1.00 (reference)
Part-time	1.16 (1.13, 1.19)	1.16 (1.05, 1.29)	1.37 (1.04, 1.80)
Unemployed	0.95 (0.93, 0.97)	1.12 (1.02, 1.23)	1.14 (0.88, 1.48)
Not in labor force	1.20 (1.17, 1.23)	1.11 (1.01, 1.23)	1.11 (0.84, 1.45)
**Living arrangement**			
Homeless	1.00 (reference)	1.00 (reference)	1.00 (reference)
Dependent living	1.09 (1.07, 1.11)	0.90 (0.83, 0.97)	0.77 (0.64, 0.92)
Independent living	1.35 (1.33, 1.38)	1.04 (0.97, 1.11)	1.08 (0.92, 1.27)
**Health insurance**			
Private insurance	1.00 (reference)	1.00 (reference)	1.00 (reference)
Medicaid	1.52 (1.47, 1.57)	1.14 (1.00, 1.30)	1.21 (0.88, 1.66)
Medicare	1.08 (1.04, 1.12)	0.77 (0.66, 0.89)	1.09 (0.74, 1.61)
None	1.56 (1.51, 1.61)	1.49 (1.32, 1.68)	1.46 (1.09, 1.94)
**Payment source**			
Self-pay	1.00 (reference)	1.00 (reference)	1.00 (reference)
Private insurance	1.78 (1.71, 1.84)	1.14 (0.99, 1.32)	0.86 (0.61, 1.22)
Medicare	1.67 (1.56, 1.79)	2.04 (1.48, 2.80)	1.36 (0.58, 3.19)
Medicaid	1.28 (1.24, 1.31)	1.16 (1.04, 1.30)	0.95 (0.74, 1.21)
Other government payments	1.22 (1.19, 1.25)	0.90 (0.82, 0.99)	0.72 (0.58, 0.89)
No charge	2.64 (2.56, 2.73)	1.59 (1.40, 1.81)	0.93 (0.67, 1.29)
Other	2.02 (1.97, 2.08)	1.89 (1.68, 2.12)	1.47 (1.10, 1.98)
**Veteran status**			
Yes	1.00 (reference)	1.00 (reference)	1.00 (reference)
No	1.03 (1.00, 1.07)	1.01 (0.86, 1.17)	1.13 (0.74, 1.72)

**Table 5 healthcare-13-00369-t005:** Adjusted odds ratio and 95% confidence interval for different patients grouped by prior treatments.

Variable	No Prior Treatment	>0 Prior Treatment
**Age**		
Age 12–17	1.00 (reference)	1.00 (reference)
Age 18–29	0.95 (0.91, 1.00)	1.08 (1.03, 1.14)
Age 30–49	0.85 (0.81, 0.89)	0.94 (0.89, 0.99)
Age 50–64	0.74 (0.70, 0.79)	0.72 (0.68, 0.76)
Age 65+	0.67 (0.60, 0.75)	0.59 (0.54, 0.64)
**Gender**		
Male	1.00 (reference)	1.00 (reference)
Female	0.97 (0.95, 0.99)	1.08 (1.06, 1.09)
**Race**		
Asian	1.00 (reference)	1.00 (reference)
Black or African American	1.74 (1.56, 1.93)	1.44 (1.30, 1.59)
Native Hawaiian/American Indian/Pacific Islander	0.84 (0.75, 0.94)	0.61 (0.54, 0.67)
Other single race	0.69 (0.61, 0.78)	1.07 (0.95, 1.20)
Two or more races	1.20 (1.06, 1.37)	1.22 (1.09, 1.36)
White	1.11 (1.00, 1.23)	1.27 (1.15, 1.40)
**Marital status**		
Never married	1.00 (reference)	1.00 (reference)
Married	0.97 (0.94, 1.00)	1.01 (0.98, 1.03)
Separated	0.89 (0.86, 0.93)	1.00 (0.97, 1.03)
Divorced	0.93 (0.90, 0.97)	0.95 (0.93, 0.97)
**Education**		
Less than Grade 8	1.00 (reference)	1.00 (reference)
Grades 9 to 11	1.25 (1.19, 1.30)	1.20 (1.16, 1.25)
Grade 12 (or GED)	1.15 (1.10, 1.20)	1.08 (1.04, 1.11)
1–3 years of college	1.09 (1.03, 1.14)	1.05 (1.01, 1.09)
4 years of college, BA/BS	0.96 (0.90, 1.02)	0.91 (0.87, 0.95)
**Source of income**		
Wages	1.00 (reference)	1.00 (reference)
Public assistance	1.82 (1.72, 1.92)	1.46 (1.41, 1.52)
Retirement/pension/disability	1.14 (1.08, 1.21)	1.15 (1.11, 1.20)
Other	1.14 (1.10, 1.20)	1.25 (1.21, 1.29)
None	1.08 (1.04, 1.12)	1.04 (1.02, 1.07)
**Employment**		
Full-time	1.00 (reference)	1.00 (reference)
Part-time	1.17 (1.12, 1.22)	1.14 (1.11, 1.17)
Unemployed	0.91 (0.88, 0.95)	0.91 (0.89, 0.94)
Not in labor force	1.23 (1.18, 1.29)	1.12 (1.08, 1.15)
**Living arrangement**		
Homeless	1.00 (reference)	1.00 (reference)
Dependent living	0.84 (0.81, 0.87)	1.14 (1.12, 1.17)
Independent living	1.03 (0.99, 1.06)	1.47 (1.44, 1.50)
**Health insurance**		
Private insurance	1.00 (reference)	1.00 (reference)
Medicaid	1.13 (1.06, 1.19)	1.69 (1.62, 1.76)
Medicare	1.09 (1.03, 1.16)	1.17 (1.12, 1.23)
None	1.46 (1.39, 1.53)	1.60 (1.54, 1.66)
**Payment source**		
Self-pay	1.00 (reference)	1.00 (reference)
Private insurance	1.55 (1.46, 1.65)	1.62 (1.55, 1.70)
Medicare	1.53 (1.34, 1.75)	1.49 (1.37, 1.62)
Medicaid	1.40 (1.32, 1.47)	1.15 (1.11, 1.19)
Other government payments	1.23 (1.18, 1.29)	1.11 (1.08, 1.14)
No charge	3.02 (2.84, 3.21)	2.26 (2.17, 2.34)
Other	1.41 (1.33, 1.50)	1.83 (1.76, 1.89)
**Veteran status**		
Yes	1.00 (reference)	1.00 (reference)
No	1.05 (0.98, 1.13)	1.05 (1.01, 1.09)

## Data Availability

Data is publicly available.

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
