# Peer review of "The Relationship of Socioeconomic Factors and Substance Abuse Treatment Dropout"

_healthcare, 2025, doi:10.3390/healthcare13040369_

Round 1

Reviewer 1 Report (Previous Reviewer 3)

Comments and Suggestions for Authors

Dear, as already indicated in the email I reproduce here the same comment I made as requested by Becka Song. The manuscript is complete, in all parts, and suitable for publication.

Author Response

Thank you for your positive feedback, we appreciate it. 

Reviewer 2 Report (New Reviewer)

Comments and Suggestions for Authors

Dear Authors,

Please, see below few comments you might consider or clarify

Explain why certain discharge categories (e.g., “other”) were excluded and discuss potential biases this might introduce.

Provide more detail on how missing data were handled (e.g., whether imputation or case-wise deletion was used).

Justify the grouping of age and race categories, especially when combining small groups, as this may affect interpretability.

Include a brief justification for the use of Cramer’s V for correlation analysis and why it is appropriate for this study.

Avoid overstating causal implications; explicitly state that the study identifies associations due to its cross-sectional design.

Expand on the role of non-socioeconomic factors (e.g., mental health comorbidities, treatment quality) that may contribute to dropout but were not examined in this study.

Discuss how geographic disparities, such as differences in rural vs. urban access to care, may interact with socioeconomic factors.

Best wishes

Author Response

Comment 1: Explain why certain discharge categories (e.g., “other”) were excluded and discuss potential biases this might introduce.

Response 1: The population of discharge reason being “Other” is 4%. Since it is relatively small, we consider it does not cause much bias in the analysis. [Text added in Section 2.2]

Comment 2: Provide more detail on how missing data were handled (e.g., whether imputation or case-wise deletion was used).

Response 2: Casewise deletion was utilized to remove these records with "other" or “unknown” categories. [Please see edits to Section 2.2 and 2.3.]

Comment 3: Justify the grouping of age and race categories, especially when combining small groups, as this may affect interpretability.

Response 3: Age 12-14 and age 15-17 are grouped into age 12-17 to better represent the teenager age group. Age 18-20, 21-24, 25-29 are grouped as age 18-29 to avoid low record counts in granular groups, and age 18-29 can also better represent emerging adult group. Age 30-34, 35-39, 40-44, 45-49 are grouped as age 30-49 represents early middle adulthood which is the period people start families, careers and relationships. Age 50-54, 55-64 are grouped as age 50-64 to represent middle adulthood which is the period of time of career peak and caring for families. Age over 65 is kept as its own age group to represent senior adulthood or senior people when they reach retirement. Indigenous groups e.g. Native Hawaiian, American Indian, Pacific Islander are grouped as one race group since indigenous groups have their own unique languages, cultures, and spiritual beliefs and distinguish themselves from other races. [Please see edits to section 2.3]

Comment 4: Include a brief justification for the use of Cramer’s V for correlation analysis and why it is appropriate for this study.

Response 4: Cramer’s V is the most commonly used strength test for Chi-square independence measure. It shows how strongly two categorical fields are associated, making it ideal for our study. [Text and citation included in main text Section 2.4.] 

Comment 5: Avoid overstating causal implications; explicitly state that the study identifies associations due to its cross-sectional design.

Response 5: We have revised some languages in our text to emphasize the association. See changes in red.

Comment 6: Expand on the role of non-socioeconomic factors (e.g., mental health comorbidities, treatment quality) that may contribute to dropout but were not examined in this study.

Response 6: The treatment environment, for example, whether adjustments can be made to accommodate specific patient needs, is important to the patient's decision to retain in a treatment. Poorer psychiatric functioning, anxiety sensitivity, and having personality disorders are related to dropout risks, too. [This text was added into our manuscript Section 4.]

Comment 7: Discuss how geographic disparities, such as differences in rural vs. urban access to care, may interact with socioeconomic factors.”

Response 7: Urban area and rural area may have different access to treatment. Compared to urban areas, people living in rural areas generally have significantly less access to healthcare and treatments due to a shortage of treatment facilities, longer travel distances to reach medical facilities, and limited availability, often leading to delayed or forgone medical treatment; this can result in poorer health outcomes for rural populations. [Please see edits in Section 4 paragraph 3.]

Round 2

Reviewer 2 Report (New Reviewer)

Comments and Suggestions for Authors

Thank you for addressing the comments 

This manuscript is a resubmission of an earlier submission. The following is a list of the peer review reports and author responses from that submission.

Round 1

Reviewer 1 Report

Comments and Suggestions for Authors

This article aims to investigate the risk of dropout from substance use disorder programs. While the research topic is undoubtedly interesting and the analyzed data is robust and appropriate for the study's objectives, the paper has several significant shortcomings.

First, the introduction is largely a brief listing of other articles, lacking a coherent narrative or critical synthesis of the findings presented.

The description of the data source is too concise to provide adequate clarity. Several questions arise during the reading that remain unanswered. For instance, the statement "patients who were admitted more than once are counted multiple times" raises concerns. Does this imply that there is no method to control for multiple entries of the same individual in the dataset? If so, this poses a serious issue. Perhaps this could be addressed by focusing on a single cohort rather than spanning four years, especially given the nearly one million patients included, which results in a highly inflated dataset (i.e., leading to inflated confidence intervals where many results appear statistically significant, even those with very small odds ratios).

Additionally, the number of sociodemographic characteristics analyzed is excessively large, complicating the interpretability of the results without addressing the implications of this (non)selection. For example, to what extent could there be multicollinearity issues between race and ethnicity? What insights does maintaining such granularity in the categorization of these variables provide, if any?

Another issue arises at the beginning of the results section, where the reader is informed that "core-based statistical area (CBSA) is highly correlated with treatment dropout." However, this variable is introduced without prior context.

The section on descriptive statistics is frustrating, as it merely reiterates values already presented in Table 2, lacking any critical synthesis of this information.

Furthermore, Paragraph 3.3 is simply a list of results found in Table 3, presenting an additional challenge. If patients aged 12-17 are your reference category in the regression, how can you derive their odds ratio? What is the reference point for interpreting that same odds ratio?

Finally, the article appears to misuse the strategy of subsampling data based on specific characteristics. These factors should be included in a primary regression model, controlled through interaction with variables that have a strong theoretical basis for influencing behavior. The subsampling approach used is neither clear nor adequately justified in the methodology section.

The findings section seems more focused on aligning the results with previous research than on critically presenting and interpreting the authors’ own results.

Reviewer 2 Report

Comments and Suggestions for Authors

Title is acceptable

Edit highlights All highlights points include many words

Abstract: edit Results subsection ( mentioned the findings well)

introduction section is well written, has good citations and mentioned the study problem well.

Methodology: Add ethical approval for the study

Results: Can you add at least one figure to mention any correlation between gender and SUDs.

Discussion: Try to cite articles from this journal according to their availability.

Conclusion extracted from findings

Reviewer 3 Report

Comments and Suggestions for Authors

Dear authors, I am well pleased to be able to review your work, which I predict is in my opinion excellent! It is clearly, comprehensively, and linguistically correct as well; the references are appropriate and centered. The abstract is comprehensive, as is the introduction which is able to introduce the topic of study in a straightforward and unexcessive manner. Excellent insight to include highlitghts as well. The materials and methods are appropriate and functional for the research objectives, as are the study variables and the statistical analysis carried out, however, I would appreciate a specific paragraph on the research objectives, unbundled from the introduction and methods (or at least subsequent to the methods section). The results are clear and specific, and the discussions are complementary and complete with respect to the research data; however, I would suggest including a table of the study's strengths and weaknesses in the discussions section. The conclusions are consistent with the work presented. I suggest minimal revisions in relation to the critical issues raised. Good work!